Using Bidirectional Encoder Representations from Transformers (BERT) to predict criminal charges and sentences from Taiwanese court judgments

http://orcid.org/0009-0007-4516-2256 Peng Yi-Ting d07921014@g.ntu.edu.tw
Lei Chin-Laung
Department of Electrical Engineering, National Taiwan University , Taipei City , Taiwan
Brunello Andrea
Electronic publication date: 2024 Jan 31
Publication date: 2024
Volume: 10
Electronic Location ID: e1841
Received 2023 Aug 2; Accepted 2024 Jan 9
Copyright: © 2024 Peng and Lei
Copyright year: 2024
Copyright holder: Peng and Lei
License: This is an open access article distributed under the terms of the Creative Commons Attribution License, which permits unrestricted use, distribution, reproduction and adaptation in any medium and for any purpose provided that it is properly attributed. For attribution, the original author(s), title, publication source (PeerJ Computer Science) and either DOI or URL of the article must be cited.
License URL: https://creativecommons.org/licenses/by/4.0/

Keywords: National language processing, Bidirectional encoder representations from transformer (BERT), Legal artificial intelligence (Legal AI), Artificial intelligence law (AI Law), Legal judgment prediction

Funding: The authors received no funding for this work.

==============================
People unfamiliar with the law may not know what kind of behavior is considered criminal behavior or the lengths of sentences tied to those behaviors. This study used criminal judgments from the district court in Taiwan to predict the type of crime and sentence length that would be determined. This study pioneers using Taiwanese criminal judgments as a dataset and proposes improvements based on Bidirectional Encoder Representations from Transformers (BERT). This study is divided into two parts: criminal charges prediction and sentence prediction. Injury and public endangerment judgments were used as training data to predict sentences. This study also proposes an effective solution to BERT’s 512-token limit. The results show that using the BERT model to train Taiwanese criminal judgments is feasible. Accuracy reached 98.95% in predicting criminal charges and 72.37% in predicting the sentence in injury trials, and 80.93% in predicting the sentence in public endangerment trials.

Introduction

Natural language processing (NLP) has been applied in many fields, such as finance and healthcare, and can also be used in the legal field. Natural language processing can be classified into two types: Natural Language Understanding (NLU) and Natural Language Generation (NLG). The development of natural language processing has shifted from early word-embedding techniques to the prevalent use of pre-trained language models since 2018. Its applications include machine translation, text categorization, information extraction, summarization, dialogue systems, and medicine (Khurana et al., 2023). There are several types of natural language processing language models. Examples include Bidirectional Encoder Representations from Transformers (BERT) (Devlin et al., 2019), Universal Language Model Fine-Tuning (ULMFiT) and XLNet (Yang et al., 2020). These language models can be used for sentiment analysis, text classification, and question answering (Howard & Ruder, 2018). The generative language model has gained recent popularity with the introduction of GPT-3 and GPT-4 (Bubeck et al., 2019).

Taiwan is a civil law country whose judicial system primarily comprises judges, prosecutors, and lawyers in court trials (Kennedy, 2007). In the 1990s, proposals were made for significant changes to Japan’s criminal justice system, one of which created the lay judge system. The lay judge system aimed to increase public trust (Reichel & Suzuki, 2015). A 2016 Japanese survey showed a 75.6% trust rate in judges. A 2015 survey done by the National Chung Cheng University in Taiwan, however, showed an 84.6% distrust rate for judges. This distrust came to a head with two court cases in 2010: a child sexual abuse case followed by the Taiwan Supreme Court overturning a guilty verdict related to the sexual abuse of a three-year-old girl. These two cases sparked a public outcry, with the media characterizing the judges as “dinosaur judges.” The Judicial Yuan of Taiwan introduced lay judges to help rebuild public trust. This trial procedure involves six lay judges and three professional judges in Taiwan’s lay judgment system. As shown in Table 1, at least six judges must agree upon a guilty verdict. Only after this does the case proceed to the sentencing trial. If the judge decides to impose the death penalty, at least six judges must concur with the verdict; for a fixed-term imprisonment, at least five judges must concur. Concurrence from both professional and lay judges is required in both scenarios. Great expectations surround the lay judge system and the transition to the new system requires training and sentencing guidance for nonprofessional judges and nonlegal experts to ensure a successful implementation (Chin, 2019; Su, 2017). Lay judges come from various professions, giving the lay judge system the advantage of not relying solely on legal professionals to determine verdicts, resulting in fairer and more objective trials. However, the lay judge system does have potential downsides. It may make it more difficult to compare the severity of sentences in judicial judgments. For example, people from different professions may have different standards for sentencing theft.

Table 1 Rules for lay judges in Taiwanese trials.

Type	Number of judges needed with agreeing votes	Limit	
Conviction	6 or more	Consent from both judges and lay judges is require.	
Fixed-term imprisonment	5 or more	
Death penalty	6 or more	

This study used past judgments as data and designed a system that uses the text of the judgment and deep learning technology to infer the charges and sentencing degree of cases that have already been determined. This study aimed to design a system that can provide references for non-legal professionals and lay judges in the judicial process. The facts of the case still need to be determined in court through the arguments of the prosecutor and defense counsel. However, once the facts of the case are determined, they can be inputted into this system, which then uses historical case law to infer appropriate charges and sentencing. The architecture concept proposed in this study is shown in Fig. 1.

Figure 1 Concept architecture diagram.

This study explores the application of natural language processing in Taiwan’s judicial system. Publicly-available court judgments were collected and fine-tuned, using the BERT model to determine whether accurate judgments of criminal charges could be achieved. The model was then used to predict the sentence length.

The 512-token length limitation of the original BERT model needed to be revised for training on legal judgments, which often consist of thousands of words. Recent models have been introduced that expand the 512-token limit, such as the Big Bird (Zaheer et al., 2020), REFORMER (Kitaev, Kaiser & Levskaya, 2020), and Linformer (Wang et al., 2020). This study proposes a language model based on the BERT model to increase the length input text to 1,024 and 1,536 characters, respectively, to solve the 512-token limitation. Extending the input text length meant the BERT model could be trained with up to 80.93% accuracy in predicting sentence length of public endangerment cases.

There is currently no relevant literature using Taiwanese legal judgments as a dataset on BERT. This study demonstrates the feasibility of training the BERT model on these legal judgments. The system proposed in this study could provide lay judges with a reference for judgment in court cases. The proposed method can also be applied to law-related research and applications in other countries.

Related work

Legal artificial intelligence

Previous, research in natural language processing often focused on word segmentation and POS tagging (Shao, Hardmeier & Nivre, 2018). After the first International Conference on Artificial Intelligence and Law (ICAIL) held in 1987, Artificial Intelligence and Law emerged as a distinct study area (Governatori et al., 2022).

Legal Artificial Intelligence (Legal AI) has rapidly developed in recent years. Legal AI can be divided into three categories by intended user: administrators of law (e.g., judges, legislators), practitioners of law (e.g., attorneys), and those who are governed by law (e.g., people; Surden, 2019). An attention-based neural network framework can be used to predict charges in legal texts. Previous studies have shown that the neural network method is more effective than the traditional support vector machine (SVM) method for predicting charges (Luo et al., 2017). Another previous study used deep learning in the legal domain in a method named Law2Vec. This method used pre-trained legal word embeddings using the word2vec model and publicly shared domain-specific legal word embedding (Chalkidis & Kampas, 2019). Symbol-based methods and embedding-based methods have also been proposed in Legal AI for finding similar court cases for judgment predictions (Zhong et al., 2020).

A 2023 study (Katz et al., 2023) proposed dividing the task classification of legal natural language processing into seven categories: machine summarization, pre-processing, classification, information retrieval, information extraction, text generation, and resources. Recent advancements in deep learning have led to more research being done using the classification task. This study uses legal facts to classify possible charges and sentences.

General databases are difficult to apply to legal AI because law requires domain expertise, so building a legal database is very important for legal AI research. Some legal datasets can be used for legal AI research. For example, Chalkidis et al. (2019) proposed the “EURLEX57K” dataset, which includes 57,000 English documents on EU legislation. Xiao et al., 2018 proposed the first large-scale Chinese legal dataset for judgment prediction in 2018, which used over 2.6 million criminal cases published by the Supreme People’s Court of China. Zheng et al. (2021) proposed the “CaseHOLD” (Case Holdings On Legal Decisions) dataset, which comprised over 53,000+ multiple choice questions to identify the relevant holding of a cited case. Paul, Goyal & Ghosh (2022) proposed Legal Statute Identification (LSI), constructing a large dataset called the Indian Legal Statute Identification (ILSI) dataset from case documents and statutes from the Indian judiciary. Unfortunately, in Taiwan, there is no relevant legal dataset and no public legal training model available. This study obtained previous legal judgments published by Taiwan’s Judicial Yuan and then sorted and classified these judgments to build the judicial judgment database.

Legal judgment prediction (LJP)

The development of deep learning led to its application in Legal Judgment Prediction (LJP). In 2019, Yang et al. (2019) proposed a multi-perspective bi-feedback network using the word collocation attention mechanism based on the topology structure among subtasks. In 2020, Zhong et al. (2020) used a question-answering task to improve the interpretability of this network. The method also uses reinforcement learning to minimize questions (Zhong et al., 2020). In 2021, Gan et al. (2021) proposed a method integrating logic rules into a co-attention network-based model (Gan et al., 2021). In 2022, Feng, Li & Ng (2022) proposed an Event-based Prediction Model (EPM) with constraints.

Because legal judgments are very lengthy, automatic summarization of legal documents could save legal practitioners a significant amount of time. The lack of expert-annotated summaries makes it challenging to develop automated systems to help paralegals, attorneys, and other legal professionals. Agarwal, Xu & Grabmair (2022) proposed a method for extractive summarization of legal decisions that can use limited expert-annotated material in low-resource settings. In the same year, Santosh et al. (2022) and Santosh, Ichim & Grabmair (2023) used domain expertise to identify statistically predictive information and exclude legally irrelevant information.

The LJP research area is closely related to NLP, which is related to language models because of deep learning. In 2019, Devlin et al. (2019) published Bidirectional Encoder Representations from Transformers (BERT). BERT is a language model based on the transformer architecture. Many LJP researchers have applied BERT in their work. Other researchers have used the BERT Model as a base and modified it, such as the Robustly Optimized BERT Pretraining Approach (ROBERTa; Lewis & Stoyanov, 2020), A Lite BERT (ALBERT; Liu et al., 2020), ELECTRA (Clark et al., 2020), and XLNet (Yang et al., 2020). XLNet uses Transformer-XL as its pretraining framework. There is also a cybersecurity feature claims classifier based on BERT called CyBERT (Ameri et al., 2021). Chalkidis et al. (2020) proposed a legal language model called LEGAL-BERT in 2020. Table 2 shows recent articles in the LJP field. This study compares the proposed method outlined in this article with BERT and XLNet.

Table 2 Recent research articles in the LJP field.

Number	Proposed method	Legal field	The year of publication	
[1]	Using BERT to identify industry requirements for cybersecurity.	Cybersecurity claim documentation	2021	
[2]	LeSICiN: using the attribute encoder and the structural encoder to generate representations	Indian criminal law	2022	
[3]	An attention-based neural network	Chinese criminal law	2017	
[4]	Using domain expertise to strategically predict irrelevant information	European court of human rights	2022	
[5]	A modified version of the enhanced sequential inference model (ESIM) incorporating conditional encoding	European court of human rights	2023	
Notes:

CyBERT: cybersecurity claim classification by fine-tuning the BERT language model.

LeSICiN: a heterogeneous graph-based approach for automatic legal statute identification from Indian legal documents.

Learning to predict charges for criminal cases with legal basis.

Deconfounding legal judgment prediction for European Court of Human Rights cases towards better alignment with experts.

Zero-shot transfer of article-aware legal outcome classification for the European Court of Human Rights cases.

Materials and Methods

Data preparation

This study collected judgments published by Taiwan’s Judicial Yuan, which required initial data preprocessing. Because the format of judgments does not have a standard format for criminal sentences, data preprocessing was performed to extract the criminal charges, sentences, and criminal fact description before deep learning and natural language processing were performed. During this process, three key columns were extracted from the dataset for use: “Charge,” “Sentence,” and “Main Text of the Judgment.” The three columns are described as follows:

1. The “Charge” column outlines the specific criminal charges.

2. The “Sentence” column includes the imposed punishment.

3. The “Main Text of the Judgment” column provides a detailed account of the criminal facts and legal proceedings.

The resulting dataset is compared to the CAIL (Xiao et al., 2018), ECHR (Chalkidis, Androutsopoulos & Aletras, 2019), and Indian Legal Statute Identification (ILSI; Paul, Goyal & Ghosh, 2022) datasets in Table 3. As shown in Fig. 2, 85.43% of the judgments were over 512 tokens in length, with many including thousands or even tens of thousands of words, so the original BERT method, which has a 512-token limit, resulted in overfitting issues. Therefore, this study also proposes a method to extract more text from court judgments to overcome the 512-token limit of the original BERT model.

Table 3 A comparison of the study dataset and three other datasets (CAIL, ECHR and ILSI).

Dataset	This study	CAIL	ECHR	ILSI	
Language	Chinese	Chinese	English	English	
Number of judgements/cases	218,120	2,676,075	11,478	66,090	
Act	Criminal Law in Taiwan	Criminal Law in China	European Convention on Human Rights	Criminal Law in Indian	
The year of publication	2024	2018	2019	2022	

Figure 2 The content length of each judgment in the dataset.

Figure 3 describes the judgment data collected from this research, with complete descriptions of the many columns included in these judgments shown in Table 4. This study focused on “injury” and “public endangerment” cases to predict legal sentencing. It is important to note that these two crimes are not subject to the death penalty under Taiwan’s criminal law. Therefore, only judgments of innocence and fixed-term imprisonment sentences were retained in the “Sentence” column. Because there were differing judicial considerations in sentences that included fines compared to cases of simple fixed-term imprisonment sentences, cases where the imposed punishment could be commuted to a fine or additional fixed-term imprisonment with a fine were excluded. This data cleaning process was performed to enhance the accuracy of the training data.

Figure 3 The dataset example of this research.

Table 4 Complete descriptions of the columns included in the Taiwanese judgment public information collection.

Items	Description	
JID	Judgment case ID information	
JYEAR	Year of the case	
JCASE	Type of case	
JNO	Number of this case type for this year	
JDATE	Judgment date	
JTITLE	Syllabus	
JFULL	The main content of the judgment, including the plaintiff, defendant, judge, judgment result, and reason for the judgment	
JPDF	Link to download the judgment PDF	

This study was divided into two experiments: the criminal charge prediction model and the sentence prediction model. These two discriminative models were trained on the “Charge,” “Sentence,” and “Main Text of the Judgment” columns to predict criminal charges and sentences, respectively.

Performance metrics

There are several common metrics for measuring the performance of classification algorithms, including: accuracy, precision, recall, and F1, confusion matrix, and ROC-AUC score (Vakili, Ghamsari & Rezaei, 2020). This study used accuracy, precision, recall, and F1 as the evaluation metrics of algorithm performance.

Model architecture

This study used the Chunking BERT model, which calculates two sets of loss. The original judgment text was split into: two paragraphs of 512 tokens, using the first 1,024 token of the judgment (shown in Fig. 4), and three paragraphs of 512 tokens, using the first 1,536 tokens of the judgment (shown in Fig. 5). These two chunking methods were then compared, and the results showed that more text data from the judgment did not necessarily lead to better performance.

Figure 4 Splitting the original judgment text into two paragraphs of 512 tokens each.

Figure 5 Splitting the original judgment text into three paragraphs of 512 tokens each.

Each judgment contained words that were irrelevant to case sentencing, such as addresses and personal names. The severity of criminal sentencing is primarily determined by the criminal facts, circumstances, methods, and seriousness of their impact (Welleck et al., 2019). A flowchart of the proposed method is shown in Fig. 6. The main text of the judgment was first extracted and then used as input for the language model training process. The main text of each judgment was then segmented into two sets of 512 tokens, which were then converted into embeddings using the BERT model, resulting in the creation of chunk A and chunk B. During the training of the language model, the loss values generated by training these two chunks were averaged, and the softmax function was applied to generate the predicted category. This process was then repeated, instead using three sets of 512 tokens from each judgment to train the language model.

Figure 6 The flowchart of the proposed method.

Criminal charge prediction

The ten most frequently occurring types of crimes were selected from the dataset of judgments: injury, theft, obstructing sexual autonomy, public endangerment, fraud, defamation, negligent injury, gambling, forgery, and negligent homicide. The BERT model was then used to train the criminal charge predictions of these ten types of crimes. As shown in Fig. 7, there were no training issues encountered in criminal charge prediction, indicating that using court judgments as a dataset for the BERT classification task can successfully classify criminal charges.

Figure 7 Accuracy (A) and loss (B) value of predicting the criminal charges based on the original BERT model.

Sentence prediction

BERT was also used as the model for predicting sentences. Two types of crimes—injury and public endangerment—were chosen for the sentence prediction model, as these two datasets were relatively complete, with 15,143 and 88,634 cases, respectively.

However, these datasets could not be directly used to train the sentence prediction model because many sentence lengths for crimes in the crime classification dataset were mixed with indefinite sentence lengths or other complex factors. For example, the defendant may have been sentenced to 1 month in prison, with a two-month probation period, or may have been sentenced to 2 months imprisonment with a fine of 50,000 New Taiwan Dollars (NTD).

This study only used data with simple categories of not guilty or imprisonment without suspension to avoid the influence of other complex factors on the model training results. Data with suspended sentences or fines were also excluded. After filtering the datasets based on these criteria, 9,103 injury cases and 83,550 public endangerment cases were included for training sentence prediction.

As shown in Table 5, injury cases were classified into two categories. We assign two types of labels to each injury case. Label 0 signifies that the penalty for the injury case is innocent or less than 2 months of imprisonment. There are a total of 5,787 cases under this category. Label 1 represents a penalty for injury of over 2 months imprisonment, with 3,316 cases: innocent to a sentence of less than 2 months imprisonment, which accounted for 63.57% of the included injury cases, and a sentence of over 2 months of imprisonment, which accounted for 36.43% of the included injury cases.

Table 5 Injury sentence categories.

Category ID	Penalty	Number of cases	
0	Acquittal to less than 2 months imprisonment	5,787	
1	Over 2 months imprisonment	3,316	
	Total	9,103	

As shown in Table 6, public endangerment trials were classified into three categories. We assign three types of labels to each public endangerment case. Label 0 signifies that the penalty for the public endangerment case is innocent or less than 4 months of imprisonment. There are a total of 47,661 cases under this category. Label 1 represents a penalty for public endangerment ranging from 4 months to under 6 months of imprisonment, with 22,503 cases. Label 2 indicates a public endangerment penalty of over 6 months of imprisonment, encompassing 13,386 cases: innocent to a sentence of less than 4 months imprisonment, which accounted for 57.04% of included public endangerment cases, 4 months to less than 6 months of imprisonment, which accounted for 26.93%, and over 6 months of imprisonment, which accounted for 16.03% of the included public endangerment cases.

Table 6 Public endangerment sentence categories.

Category ID	Penalty	Number of cases	
0	Acquittal to less than 4 months imprisonment	47,661	
1	4 months imprisonment to less than 6 months imprisonment	22,503	
2	Over 6 months imprisonment	13,386	
	Total	83,550	

The Injury cases were used in BERT to train sentencing prediction, using the first 512 tokens based on BERT’s 512-token limit. Figure 8 shows that after training with batch sizes of eight and 40 epochs, the validation accuracy remained at 68% to 70% without a significant improvement, and the validation loss continued to increase, indicating a significant overfitting problem. This is likely because the text of the judgments contained thousands of characters, and using only the first 512 tokens made it difficult for BERT to find correlations, leading to overfitting. These results indicate that for Taiwanese judgments, the original BERT model cannot be used to predict criminal sentences. The following sections describe the methods proposed in this study.

Figure 8 Accuracy (A) and loss (B) curves of predicting sentence in injury cases using the original BERT model.

Public endangerment cases were then used in BERT to train sentence prediction to verify this finding. The accuracy and loss curves of predicting public endangerment sentences on BERT are shown in Fig. 9. These curves exhibited a similar trend as those trained on injury cases, with overfitting being a common issue. Because overfitting still occurred when using the public endangerment dataset, which contained 83,550 cases making it approximately 9.17 times larger than the injury dataset, insufficient data is likely not the leading cause of the overfitting.

Figure 9 Accuracy (A) and loss (B) curves of predicting sentence in public endangerment cases using the original BERT model.

These experiments show that using judgment data in BERT for criminal sentencing prediction can easily lead to overfitting problems, and the accuracy of the validation data cannot be effectively improved, causing the model to be inaccurate. Because this may be because of BERT’s 512-token limit, a method was proposed to chunk the input characters and increase the input to two or three sets of 512 tokens each. Because the input characters include the CLS and SEP tokens, the actual data inputted is 510 characters, taking the average loss values generated for each set of 512 tokens.

Results

Using the original BERT method to determine the type of crime, or case charges, resulted in an accuracy level of 97.53%. However, the original BERT method resulted in low model accuracy and overfitting issues when predicting the criminal sentence. Considering the cost of training time, follow-up experiments were conducted using 1,024- and 1,536-word chunks, referred to as the “two-chunking BERT model” and the “three-chunking BERT model,” respectively.

Data sets

The source data were public court judgments made by the Judicial Yuan of Taiwan. Each judgment consisted of eight columns: JID, JYEAR, JCASE, JNO, JDATE, JTITLE, JFULL, and JPDF. Complete descriptions of the content of each column are presented in Table 6. This study collected the only “JTITLE” and “JFULL” columns from each case. The criminal classification of all data in this study is shown in Table 7.

Table 7 Classification of criminal charges.

Criminal charge	Number of cases	
Injury	15,143	
Larceny	42,804	
Obstructing sexual autonomy	1,578	
Public endangerment	88,634	
Fraud	24,105	
Defamation	3,884	
Negligent injury	29,905	
Gambling	4,103	
Forgery	5,254	
Negligent homicide	2,710	
The total number of criminal charges	218,120	

The “JTITLE” column, also known as “syllabus,” contains information about the criminal charges and sentencing of the involved parties, as well as other descriptive details (i.e.: “Mr. A is charged with endangering public safety and is liable for a fine”). The experimental data are presented in Table 7. The ten most common types of crime were included in the dataset, resulting in a total dataset of 218,120 judgments. The dataset is randomly divided into training, validation, and test sets, with the training set representing 80% of the dataset. The validation and test sets each account for 10% of the dataset. In this study, original BERT, two-chunking BERT, three-chunking BERT, and XLNet language models utilize the dataset, which is randomly sampled.

Criminal charge prediction

As shown in Figs. 10 and 11, the experimental results indicate that the criminal charge prediction improved in the two-chunking BERT and three-chunking BERT model. However, compared to the original BERT, the accuracy only increased from 97.53% to 98.95%, an improvement of 1.42%.

Figure 10 Accuracy (A) and loss (B) curves of predicting criminal charges based on the two-chunking BERT model.

Figure 11 Accuracy (A) and loss (B) curves of predicting criminal charges based on the three-chunking BERT model.

After ten epochs, the accuracy of the two-chunking BERT model was 98.5%, 0.97% higher than the original BERT model. Both the two-chunking BERT and three-chunking BERT models showed an increase in accuracy over the XLNET model, which achieved an accuracy of 98.16%. Comparing the accuracy and loss values of the original BERT, two-chunking BERT, and three-chunking BERT models showed that using the chunking method significantly improved criminal charge prediction.

A comparison of the accuracy between the two-chunking BERT model and the three-chunking BERT model is shown in Table 8, with the three-chunking BERT model showing a significant improvement in accuracy. These results indicate that accurate criminal charge prediction is achievable when using the first 1,536 tokens of a judgment.

Table 8 A comparison the accuracy, precision, recall, and F1 of Criminal charge prediction using BERT and XLNET models.

Model	Accuracy	Precision	Recall	F1	
XLNET	98.16%	98.17%	98.16%	98.16%	
BERT	97.53%	97.53%	97.53%	97.53%	
Two chunking BERT	98.5%	98.5%	98.5%	98.5%	
Three chunking BERT	98.95%	98.95%	98.95%	98.95%	

Sentence prediction

The loss value curves of using injury cases in BERT to predict criminal sentences are shown in Figs. 12B and 13B. After 30 epochs, the training loss and validation loss approached each other, indicating convergence. Based on this result, 30 epochs were used for these experiments.

Figure 12 Accuracy (A) and loss (B) curves of predicting sentence in injury cases using the two-chunking BERT model.

Figure 13 Accuracy (A) and loss (B) curves of predicting sentence in injury cases using the three-chunking BERT model.

After training for 30 epochs, the sentence prediction results for the two-chunking BERT and three-chunking BERT models both showed an accuracy rate of over 71%. Using the original BERT model led to overfitting and a low test accuracy, but these issues were significantly resolved after using the two-chunking BERT models. The accuracy and loss values of using injury and public endangerment cases for sentence prediction are shown in Table 9.

Table 9 Sentence prediction results using different models on both injury and public endangerment cases.

Crime type	Model	Accuracy	Precision	Recall	F1	
Injury	XLNET	70.14%	69.49%	69.96%	66.12%	
BERT	68.82%	68.40%	68.90%	68.58%	
Two-chunking BERT	72.37%	73.22%	72.31%	72.64%	
Three-chunking BERT	71.49%	70.35%	71.54%	69.13%	
Public endangerment	XLNET	79.06%	78.30%	79.05%	77.94%	
BERT	73.03%	75.41%	73.03%	73.61%	
Two-chunking BERT	80.16%	79.43%	80.16%	79.34%	
Three-chunking BERT	80.93%	80.53%	80.93%	80.50%	
Note:

Bold values indicate the optimal value of the injury and public endangerment evaluation index.

As shown in Tables 5 and 6, the injury judgment database included 9,103 cases, and the public endangerment judgment database included 83,550 cases. As shown in Table 9, the accuracy of the two-chunking BERT model was 72.37% on the injury data and 80.93% on the public endangerment data indicating that the two-chunking BERT method can solve the token limitation problem when training Taiwanese judgments.

Figures 14 and 15 show the results of the two-chunking BERT and the three-chunking model on the public endangerment data. Table 9 shows the results using testing data of the chunking BERT method on the public endangerment data. This model had an accuracy rate of 80.93%, which was 0.77% higher than the two-chunking BERT model. Similar results were found on the injury data, with the three-chunking BERT model achieving an accuracy rate of 71.49%, which was slightly lower than the two-chunking BERT model’s 72.37% accuracy. These results show that increasing the number of input characters from 1,024 to 1,536 in BERT did not significantly sentence prediction model performance.

Figure 14 Accuracy (A) and loss (B) curves of predicting sentence in public endangerment cases using the two-chunking BERT model.

Figure 15 Accuracy (A) and loss (B) curves of predicting sentence in public endangerment cases using the three-chunking BERT model.

Figures 16 and 17 compare the validation accuracy between two-chunking BERT and three-chunking BERT in injury and public endangerment. Figure 18 shows the accuracy and loss curves between the original BERT and the two-chunking BERT models on criminal injury cases. The yellow dashed and orange dashed lines in Fig. 18A represent the accuracy curves of the original BERT model. The graph shows that the validation accuracy curve of the original BERT model remained approximately 69%, but the training accuracy curve continued to increase, indicating potential issues.

Figure 16 The validation accuracy between the two-chunking and three-chunking BERT models in injury cases.

Figure 17 The validation accuracy between the two-chunking and three-chunking BERT models in public endangerment cases.

Figure 18 Comparison of the accuracy (A) and loss (B) curves of predicting sentence in injury cases between the original BERT model and the two-chunking BERT model.

Similarly, the validation loss of the original BERT model, represented by the orange dashed line in Fig. 18B, started to increase after approximately three epochs, indicating an overfitting problem that became more pronounced in the later stages of the training process.

Conversely, the silver and blue lines in Fig. 18B representing the training loss and validation loss of the two-chunking BERT method, respectively, converged as the number of epochs increased. Around 30 epochs, they line became increasingly close and eventually stabilized. Figure 19 shows the accuracy of the two-chunking BERT method on public endangerment cases. The accuracy of the two-chunking BERT model was slightly lower than that of the three-chunking BERT model, but the difference in the accuracy between the two methods was not significant, as shown in Figs. 16 and 17. The yellow dashed and orange dashed lines of Figs. 18A and 19A represent the accuracy curves of the original BERT model. The silver and blue lines in Figs. 18B and 19B represent the training loss and the validation loss values of the original BERT model. The yellow dashed curve generated by the original BERT method started with a relatively high training accuracy, reaching 72.87% and 79.06% at five epochs, respectively. However, as the number of epochs increased, the validation accuracy (orange dashed line) did not improve but slightly decreased, remaining at approximately 69% and 73%, respectively. However, when using a chunking BERT method, the training and validation accuracies steadily increased as the number of epochs increased. Additionally, the loss value continuously decreased until it stabilized at approximately 30 epochs. Comparing the results of injury and public endangerment cases shows that using a more extensive dataset led to better accuracy. This study found that using the chunking BERT method can effectively solve overfitting issues and improve accuracy.

Figure 19 Comparison of the accuracy (A) and loss (B) curves of predicting sentence in public endangerment cases between the original BERT model and the two-chunking BERT model.

Ethical discussion

Because the use of artificial intelligence on legal judgment prediction is a new application of AI, ethical considerations are especially important. Training large-scale datasets could lead to unintended risks, like uncontrolled language models and privacy issues. Judicial impartiality also needs to be ensured. The public judgments published by Taiwan’s Judicial Yuan anonymize all personal identifiable information, including name, address, and license plate information. Therefore, the model trained in this study based on these judgments avoid privacy concerns because the judgments included did not contain personal data.

In 2023, Zhang et al. (2023) proposed the “contrastive learning framework for legal judgment prediction” (CL4LJP) model. They pointed out that this model is not intended to replace judges but to help judges by providing relevant law articles, charges, or sentences (Zhang et al., 2023). Intelligent legal judgment prediction is still developing, and the inferred predictions may still be wrong. Court judgments, like hospital medical practices, significantly impact human rights. Therefore, the LJP is only recommended as a reference for the judge and should not be used alone to make final legal judgments. Similarly, the proposed model in this study aims to offer lay judges a reference point for judgements rather than providing them with final judgment decisions.

Conclusions

This study used judgments from the Taiwan District Court to predict criminal charges and the sentencing period. These judgments often contain thousands of characters with the longest single judgment containing 21,360 characters. Because the original BERT has a limit of 512 input tokens, excluding the CLS and SEP tokens, only 510 tokens can be input. Of the 218,120 data points in the criminal charge prediction dataset, only 31,334 (85.63%) of the judgments exceeded 512 tokens in their crime description indicating some the training model may not achieve a good classification performance if the original BERT model is used because of its 512-token limit.

This study proposed dividing the input text into either two sets of 512 tokens, called the two-chunking BERT method, or three sets of 512 tokens, called the three-chunking BERT method. These methods solve the low accuracy and overfitting problems of the original BERT method. As shown in Table 8, the XLNET and BERT models were compared. The experimental results show that the model proposed in this study can reach an accuracy rate of 98.95% in predicting criminal charges. The model reached an accuracy of 72.37% in predicting the sentence for injury cases and an accuracy of 80.93% in predicting the sentence for public endangerment cases.

This study used Taiwanese court judgments as the dataset. Some directions for research in the future are suggested as follows:

Include more types of criminal charges

Expanding criminal charges in a crime-prediction task would improve the accuracy and usefulness of the proposed method. Some criminal charges are closely related, which may be confusing the lay judges.

More detailed classification for sentence length in sentence prediction

The sentence prediction classification for injury and public endangerment cases was divided into three groups. If more court judgments were collected in the future, sentence length for criminal offenses could be classified into more specific categories.

Incorporating AI generative models for sentence prediction based on dialogueAI (Welleck et al., 2019)

DialogueAI can predict a trial based on a conversation with the prosecutor and defense. The dialogueAI system adds more functions, such as Legal AI and Legal reasoning (Krausová, 2017). Incorporating dialogueAI could lead to more accurate sentence predictions.

Increase the dataset

This study used district court judgments as data. In the future, judgments from Taiwan’s District Court Summary Court, High Court, and Supreme Court could be added to increase the dataset. Having more sources of judgment may help to train a more precise discriminative model.

Prediction of civil cases

This study focused on criminal cases, but there are also many applications of predictive legal technology in civil cases.

Apply this model in the other countries

The proposed model can be applied in other countries, particularly in common law countries because these countries adopt case law, which allows the use of past judgments to predict the sentence.

Supplemental Information

Supplemental Information 1 Bert law python code.

Supplemental Information 2 Xlnet law python code.

Additional Information and Declarations

Competing Interests

Author Contributions

Data Availability

The authors declare that they have no compting interests.

Yi-Ting Peng conceived and designed the experiments, performed the experiments, analyzed the data, performed the computation work, prepared figures and/or tables, and approved the final draft.

Chin-Laung Lei conceived and designed the experiments, authored or reviewed drafts of the article, and approved the final draft.

The following information was supplied regarding data availability:

The code is available in the Supplemental Files. The data is available at figshare: peng, yi ting (2023). criminal-charge-in-taiwan.csv. figshare. Dataset. https://doi.org/10.6084/m9.figshare.24647532.v13.

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
