# Peer review of "Using Bidirectional Encoder Representations from Transformers (BERT) to predict criminal charges and sentences from Taiwanese court judgments"

_PeerJ Computer Science, doi:10.7717/peerj-cs.1841_

## Round 0.1 · original submission · Major Revisions

While the authors' work is interesting and topical, there are several concerns raised by the reviewers that need to be addressed before the paper can be considered for publication.

For this reason, I strongly encourage the authors to take all of the reviewers' comments into serious consideration. Specifically, I recommend focusing especially on the following points:

- Literature Review: both reviewers have pointed out that the presented state of the art is currently incomplete. To offer readers a comprehensive understanding of the topic, additional effort should be invested in this section.

- Clarity: the paper would significantly benefit from a thorough proofreading. There are passages where the flow is hard to follow, and certain phrases or expressions are grammatically incorrect.

- Reproducibility: The description of the experimental workflow is missing several fundamental details. This hinders the reader's ability to fully grasp the proposed methodology and, more importantly, to evaluate its performance and generalizability accurately. Concerning this point, also sharing the data and code would greatly enhance the reproducibility of the study.

- Ethics: Given that the data pertains to a legally sensitive domain, it is essential to understand how the authors have addressed ethical considerations.

**Language Note:** The review process has identified that the English language must be improved. PeerJ can provide language editing services - please contact us at copyediting@peerj.com for pricing (be sure to provide your manuscript number and title). Alternatively, you should make your own arrangements to improve the language quality and provide details in your response letter. – PeerJ Staff

Reviewer 1 ·

Basic reporting

1. Basic Reporting

This paper’s discussion is not satisfactory. For one thing, the English is weak throughout, with many grammatical errors and usage mistakes (e.g., line 95, “The crime of negligent injury may be constituted.”; line 117, “In the Legal AI domain.” For another, the literature review is highly abstract and does not reflect familiarity with the field. Papers such as Zheng et al. 2021, Henderson et al. 2022, Agarwal, Xu & Grabmair 2022 discuss in great detail the issues with the length of legal decisions for the successful use of large language models for legal tasks. And closer to their own work, the authors do not address the burgeoning literature on legal judgment prediction (see, e.g., Santosh et al. 2023, Santosh et al. 2022). The authors are apparently unaware of this work. Also, several of the figures are quite confusing. For example, Figure 4 does not explain exactly how the chunks were defined from that paragraph. Figure 5 completely excludes the chunking approach. Much more effort is required for the reader to be able to clearly follow this discussion and for it to be appropriately situated in its scholarly context.

Most importantly, the authors claim that making available the dataset of Taiwanese judicial decisions is one of their principal contributions. I may have missed it, but I do not see where they have made the source data publicly available and indeed I did not see that dataset in the support materials.

Experimental design

The conceptual justification behind the paper’s design is also highly problematic to me. The authors start from the idea that there is something wrong with judging in Taiwan. But the system they have trained depends on judgments as an input which are written long after charges are laid and by the same author who is presumably about to impose a sentence. The factual content of the documents is thus conditional on the charges themselves. So the prediction task is highly attenuated from the motivation, where nobody has yet transformed an unstructured set of facts into a charging decision. Likewise the sentence is presumably about to be announced by the author of the document being used for training. Why shouldn’t we expect the author to structure the factual discussion in a way that supports the charge they themselves are about to impose?
In a precedential system, there would be perhaps independent value in taking a judgment and extracting key details like the underlying crime, sentence imposed, etc. But as the authors themselves note, this is not true of Taiwan.

Anyhow, the authors never even explain how they ensure that the charges or sentence are not included in the training materials. Their description of their preprocessing flow suggests that it might in fact be that no effort was made to cleanse the targets from the training data; e.g., naively taking the first 512 tokens (line 219) might include an introductory statement that itself describes the charges and/or the sentence. If the machine can see the charge or the sentence why would we expect the prediction task would be difficult at all?

In short, the authors frame this as a predictive task but it seems like it’s really a kind of information retrieval or summarization task where we are trying to infer information from an incomplete document that may or may not contain the desired data. And they do not adequately motivate that summarization task with a real-world problem.
I had some more mundane experimental design questions too. It seems like 200 epochs was chosen primarily based on accuracy convergence (since training loss did not converge by 200 epochs, see fig. 11B). First, this is strange because in fig 12A, there is continued convergence until roughly 300 epochs, so it’s not clear why 200 should be the endpoint. Second, why should accuracy be used when as the authors themselves note it’s a pretty problematic performance metric?

Next, all of the principal figures are basically just training and validation loss over time. But what about performance on a true holdout set at inference time? That is, the authors don’t provide detailed results for their endpoint models.

Finally, the chunking approach seems highly unlikely to yield positive results. The authors could have explored a number of other approaches, like trying to identify the most relevant spans and then training on those.

Validity of the findings

Because of the potential data leakage mentioned above, it is very unclear to me how to interpret the claimed results. Specifically, I don't know whether the claimed out-of-sample accuracy is based on true holdouts or the validation set, and I don't know whether the authors have demonstrated that the test data has been sanitized so as to exclude the target data.

·

Basic reporting

General overview:
The manuscript titled "Using BERT to predict criminal charges and sentences from court judgments" by Yi Ting Peng and Chin-Laung Lei from the Department of Electrical Engineering, National Taiwan University, aims to utilize the BERT (Bidirectional Encoder Representations from Transformers) model to predict criminal charges and the corresponding sentences based on court judgments in Taiwan. The study is divided into two main parts: prediction of criminal charges and prediction of sentences. The authors have also addressed the challenge of the 512 token limit in the BERT model by proposing a method to extend the input text length.

Recommendations:
- Literature references: While the article references several studies, it would be beneficial to ensure that all relevant literature in the field is cited to provide comprehensive context.
- Figures, tables, raw data: The author should ensure that all figures and tables are clearly labeled and accompanied by descriptive captions. If not already included, raw data should be shared or made accessible to readers for transparency.
- Definitions and proofs: It's crucial to provide clear definitions of all terms, theorems, and methodologies used. If not already present, detailed proofs or justifications for the chosen methods should be included.

Experimental design

Recommendations:
- Ethical standards: The article should explicitly mention any ethical considerations taken during the research, especially given the sensitive nature of criminal judgments.
- Method details: The methods section should provide a detailed step-by-step description of the procedures, ensuring that another researcher could replicate the study based on the information provided.

Validity of the findings

Recommendations:
- Data transparency: It's essential to provide access to all underlying data or, at the very least, a comprehensive summary of the data. This ensures transparency and allows readers to assess the robustness of the findings.
- Controls in place: The article should detail any controls used during the research to ensure the validity of the findings. This could include information on how the data was cleaned, processed, or any biases accounted for.

Additional comments

Recommendations:

1. Clarity & Language:
The manuscript would benefit from a thorough proofreading to correct minor grammatical errors and improve sentence structures.
The term "national language processing" appears to be a typographical error and should be corrected to "natural language processing."

2. Dataset & Evaluation:
It would be beneficial if the authors provided more details about the dataset they used, including its size, nature.
A more in-depth evaluation, including comparisons with other state-of-the-art models and a discussion on potential limitations, would enhance the manuscript's quality.

3. Future Work:
The authors might consider discussing potential future directions, such as the application of their model in other legal systems or its integration with other AI techniques.

4. Other comments
Given the chunks provided, it's recommended that the entire article be reviewed to ensure all sections are covered comprehensively. This includes checking for consistency in language, ensuring all claims are supported by evidence, and that the article provides a meaningful contribution to the field.

---

## Round 0.2 · Minor Revisions

The authors have significantly enhanced the manuscript compared to its original draft. However, additional improvements are required in the experimental section. In particular:

- There is a need for a clear explanation of the methodology used to determine the training, validation, and test sets, including their alignment with those utilized in the comparative study.

- It is suggested to highlight the label distribution within the datasets, as this information is useful for the correct assessment the accuracy metric.

·

Basic reporting

I thank the authors for the great work they have done in revising their article. I have been able to see that they have not only heeded my recommendations, but have improved the manuscript as much as they could, over and above the requests I made.

I consider the article to be of high quality and interesting for this journal.

Experimental design

I don't have more comments for this section. Everything is fine.

Validity of the findings

I don't have more comments for this section. Everything is fine.

Additional comments

I don't have more comments for this section. Everything is fine.

---

## Round 0.3 · accepted · Accept

The authors have addressed all of the reviewers' comments.
The paper is now ready for publication.